# Imagine to Ensure Safety in Hierarchical Reinforcement Learning

## Abstract

This work investigates the safe exploration problem, where an agent must maximize performance while satisfying safety constraints. To address this problem, we propose a method that includes a learnable world model and two policies, a high-level policy and a low-level policy, that ensure safety at both levels. The high-level policy generates safe subgoals for the low-level policy, which progressively guide the agent towards the final goal. Through trajectory imagination, the low-level policy learns to safely reach these subgoals. The proposed method was evaluated on the standard benchmark, SafetyGym, and demonstrated superior performance quality while maintaining comparable safety violations compared to state-of-the-art approaches. In addition, we investigated an alternative implementation of safety in hierarchical reinforcement learning (HRL) algorithms using Lagrange multipliers, and demonstrated in the custom long-horizon environments SafeAntMaze that our approach achieves comparable performance while more effectively satisfying safety constraints, while the flat safe policy fails to accomplish this task.

## 1 Introduction

Exploration is one of the most critical capabilities for Reinforcement Learning (RL) agents, enabling them to discover optimal behaviors for achieving predefined goals. One of the main challenges in applying RL algorithms to the real world is the lack of safety guarantees during the exploration process, which can lead to damage to expensive hardware or dangerous situations (Ray et al., 2019). Therefore, there is an urgent need for safe reinforcement learning approaches that address the safe exploration problem, which we investigate in this work.

Most current safe reinforcement learning methods employ a Lagrangian-based approach (Huang et al., 2024; Jayant & Bhatnagar, 2022; Hogewind et al., 2023; Ha et al., 2021), where the RL controller maximizes performance while minimizing safety constraint violations through the use of a Lagrangian multiplier. A primary problem with Lagrangian-based methods, as noted in (Yu et al., 2022), is that it is a complex challenge for the RL controller to simultaneously maximize performance and ensure safety. One possible solution to this problem is to use safety layers (Roza et al., 2022), where a separate safety layer module is responsible for maintaining safety. Otherwise, we can simplify the performance maximization task for the RL controller by using a high-level policy that generates intermediate subgoals and guides the agent toward the final goal. Thus, the RL controller's task remains to maximize performance and safety over a short horizon up to each subgoal.

Model-based reinforcement learning approaches allow efficient training of a transition function that can be used both for action planning by the agent (Jayant & Bhatnagar, 2022; Liu et al., 2020) and for training the RL controller in the imagination (Janner et al., 2019; Hafner et al., 2023). The use of such a model allows the agent to reduce the execution of potentially dangerous actions during real world exploration. Risks about learning a policy are embedded in the model, while the agent needs to safely explore an environment to learn this world model.

In this work, we propose a method **ITES** (**I**magine **T**o **E**nsure **S**afety in Hierarchical Reinforcement Learning)[1] that uses a high-level policy to simplify the complex optimization task for the RL con-

---

[1]The code repository: https://anonymous.4open.science/r/ITES-677D.

troller by generating safe intermediate subgoals. We also use the world model to verify the safety of actions taken by the RL controller in the imagination before executing them directly, analogous to a Lagrangian-based approach (the scheme of our approach is presented in Figure 1).

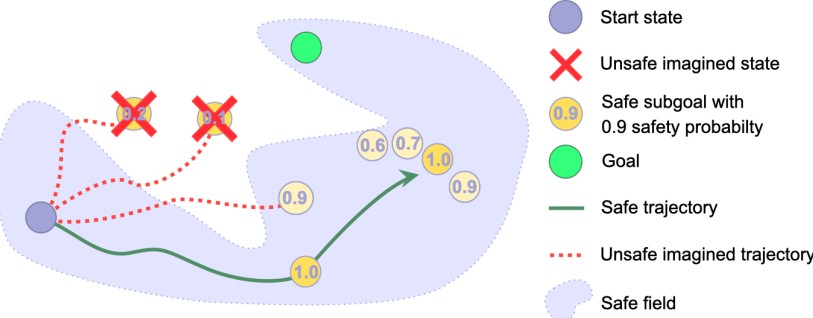

Figure 1: For a flat controller, simultaneously optimizing reward and ensuring safety during goal achievement poses a significant challenge. The ITES generates intermediate subgoals, **orange circles**, to achieve the main goal, **green circle**, which enables the controller to optimize safety over a short horizon for each of these subgoals. To ensure that the entire trajectory remains safe, the proposed method first predicts a safe subgoal for the controller, and second, the controller in imagination (via world model) optimizes safety towards that subgoal.

Our main contribution is:

- We propose using Hierarchical Reinforcement Learning for enhancing performance in Safe Reinforcement Learning.

- We propose a safe hierarchical method — using a world model to ensure the safety of the controller in an imagination, as well as a cost model for generating safe subgoals.

- A similar method for integrating safety into HRL policy based on Lagrangian multipliers has been investigated, demonstrating that our proposed method provides better safety while maintaining comparable performance.

- We demonstrate on the SafetyGym benchmark with short-horizon tasks that our method outperforms state-of-the-art methods in performance while only slightly compromising on safety. In contrast, on the long-horizon SafeAntMaze, our method surpasses both safety and performance, whereas the baselines fails to accomplish this task.

## 2 PRELIMINARIES

### 2.1 CONSTRAINED GOAL-CONDITIONED MDP

We investigate the mathematical formulation of the Safe Exploration problem introduced in Ray et al. (2019) through the framework of a Constrained Markov Decision Process (CMDP, Altman (1999)), represented as $\langle \mathbb{S}, \mathbb{A}, p, R, c, d, \mu, \gamma \rangle$. Both the state space $\mathbb{S}$ and the action space $\mathbb{A}$ are assumed to be continuous. The environment transition function $p(s'|s, a)$ specifies the probability density of reaching $s' \in \mathbb{S}$ after taking action $a \in \mathbb{A}$ in state $s \in \mathbb{S}$. The initial state distribution $\mu(s_0)$ defines the probability density of beginning an episode at state $s_0$. It is generally assumed that the agent does not know the transition dynamics $p(s'|s, a)$. For each transition $\langle s, a, s' \rangle$, the environment produces a scalar external reward $r(s, a, s')$ and another scalar $c(s, a)$ as the cost. The parameter $d \in \mathbb{R}$ represents the cost limit, indicating the maximum allowable sum of costs over an episode. The optimization problem is to maximize cumulative external rewards from an environment: $J(\pi) = \mathbb{E}_\pi \sum_t \gamma^t r(s, a, s')$, while satisfying cost constraint: $J_c(\pi) \leq d$.

Following Zhang et al. (2020a), we consider hierarchy framework to solve the problem. We adopt a hierarchical agent policy $\pi$ with two levels: a high-level controller with policy $\pi^h_{\theta_h}(s_g|s)$ and a low-level controller with policy $\pi^l_{\theta_l}(a|s, s_g)$. These controllers are parameterized by separate neural

network approximators, with parameters $\theta_h$ and $\theta_l$, respectively. The high-level controller seeks to maximize the external reward:

$$\max_{\theta_h} J_{ex}(\pi^h_{\theta_h}), \text{ where } J_{ex} = \mathbb{E}_{\pi^h_{\theta_h}} \sum_\tau R^h_\tau = \mathbb{E}_{\pi^h_{\theta_h}, \pi^l_{\theta_l}} \sum_\tau \sum_{t=k\tau}^{t=k\tau+\tau} r(s_t, a_t, s_{t+1}). \quad (1)$$

This policy generates high-level actions in the form of subgoals $s_g \sim \pi^h_\theta(s_g|s)$, where $s_g \in \mathbb{G}$, at intervals of $k$ time steps ($k > 1$ is a predefined hyperparameter). The goal space $\mathbb{G}$ is a sub-space of $\mathbb{S}$, with a known mapping function $\phi : \mathbb{S} \to \mathbb{G}$. The low-level policy performs a primary action $a \sim \pi^l_{\theta_l}(a|s, s_g), a \in \mathbb{A}$ at every time step. This policy is modulated by intrinsic rewards $r_{in}$ for reaching subgoals generated by high-level controller. The intrinsic reward $r_{in}(s, s_g)$ is a negative euclidean distance between mappings of current state $s$ and subgoal $s_g$. The low-level objective is to maximize the cumulative intrinsic rewards $J_{in}$:

$$\max_{\theta_l} J_{in}(\pi^l_{\theta_l}), \text{ where } J_{in} = \mathbb{E}_{s_g \sim \pi^h_{\theta_h}, \pi^l_{\theta_l}} \sum_{i=t}^{i=t+k} r_{in}(s_i, s_g) \text{ and } r_{in}(s_i, s_g) = -||\phi(s_i) - s_g||. \quad (2)$$

The overall objective for the agent policy is to find such parameters $\theta_l, \theta_h$ that the $J_{ex}, J_{in}$ are maximized and the cumulative cost $J_c(\pi) = \mathbb{E}_{\pi^h_{\theta_h}, \pi^l_{\theta_l}} \sum_\tau \sum_{t=k\tau}^{t=k\tau+\tau} c(s_t, a_t)$ is limited:

$$\pi^*_{\theta_h} = \arg\max J_{ex}\pi^h_{\theta_h}) \quad \pi^*_{\theta_l} = \arg\max J_{in}(\pi^l_{\theta_l}) \text{ s.t. } J_c(\pi) \leq d. \quad (3)$$

## 2.2 MODEL LEARNING

To model the transition function $p(s'|s, a)$, we selected the world model from Janner et al. (2019). The main idea is to represent the World Model $M_{\theta_m}$ as an ensemble of $n$ models, each of which takes the form $s' = M_i(s, a)$ and is a neural network. The weights ensemble is denoted as $\theta_m$. The final prediction is obtained by averaging the predictions from each of the $n$ models, which helps to reduce both epistemic and aleatoric uncertainties:

$$M_{\theta_m}(s, a) = \frac{1}{n} \sum_{i=1}^{n} M_i(s, a). \quad (4)$$

## 3 METHOD

The proposed approach consists of two components: safety of generated subgoals and safety of the RL controller. To generate subgoals within a safe region, we utilize a classifier for dangerous/safe states in the environment. In order to ensure that the agent adheres to safety constraints while achieving subgoals, we maximize imagined safety using a world model.

## 3.1 HIERARCHY STRUCTURE

As a base algorithm for our approach, we adopted HRAC (Zhang et al. (2020a), (Hierarchical Reinforcement learning with k-step Adjacency Constraint), which consists of two policies: high-level controller $\pi^h_{\theta_h}(s_g|s)$ and low-level controller $\pi^l_{\theta_l}(a|s, s_g)$. The main idea is to generate subgoals that are at a specified distance $k$ from the agent's current position. For this purpose, an Adjacency Network $\psi$ is employed, which maps each state $s$ to its emedding $\psi(s)$. This network is trained using the following loss function:

$$\hat{L}_{adj} = \mathbb{E}_{s_i, s_j \in S} \; l \cdot \max(||\psi(s_i), \psi(s_j)|| - \epsilon, 0) + (1 - l) \cdot \max(\epsilon + \delta - ||\psi(s_i), \psi(s_j)||, 0) \quad (5)$$

where $\delta > 0$ is a margin between embeddings, $\epsilon$ is a scaling factor, and $l \in \{0, 1\}$ represents the label indicating $k$-step adjacency. This loss function enables the Adjacency Network to predict embeddings for states $s_i$ and $s_j$ that are $k$ steps apart, such that the condition $||\psi(s_i) - \psi(s_j)|| < \epsilon$ is satisfied.

Subsequently, this network is utilized to update the high-level controller based on the TD3 algorithm (Fujimoto et al., 2018), incorporating a component into its loss function:

$$L^h = -Q^h(s, s_g) + \beta^h_{adj} L_{adj}(s, s_g), \tag{6}$$

here $\beta^h_{adj}$ is adjacency loss coefficient—the scaling hyperparameter. For the low-level controller, the TD3 algorithm is employed without modifications with the loss function:

$$L^l = -Q^l(s, a, s_g), \tag{7}$$

where $Q^h$ and $Q^l$ are action value functions (approximated by neural networks) which estimate discounted external and internal returns respectively.

## 3.2 SUBGOAL SAFETY: HIGH-LEVEL SAFETY

The scheme for updating the high-level policy is illustrated in Figure 2. The update is carried out through transitions $\langle s, s_g, R^h \rangle$ and modules such as the cost model $C_M$ and the $Q^h$ function, which calculate the safety and utility of the generated subgoal.

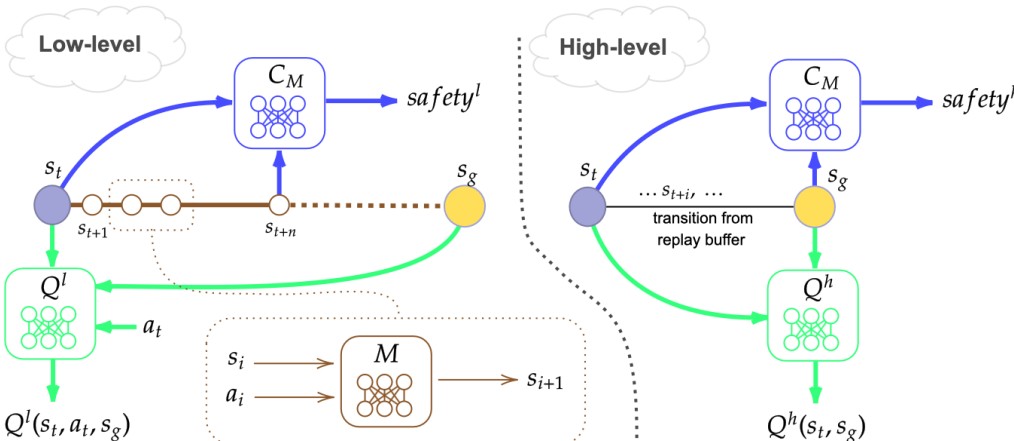

Figure 2: The scheme illustrates the training process for the low-level policy and the high-level policy. **Left**: the cost model $C_M$ and the world model $M$ are responsible for ensuring safety during the updates of the low-level policy, while the $Q^l$-function is utilized to optimize the reward. **Right:** the cost model $C_M$ is employed for safety considerations in updating the high-level policy, with the $Q^h$-function used for reward optimization.

**Cost model.** The objective of the cost model $C_M$ is to predict the probability of the safety for an arbitrary subgoal $s_g$ using local information about obstacles surrounding the agent (e.g., lidar data or images) denoted as $o_{obst}$, as well as the position of the agent denoted as $o_{pos}$ which we extract from the agent state $s$:

$$safety^h(s_g) = C_{M_{\theta_c}}(s_g, o_{obst}, o_{pos}) \in [0, 1]. \tag{8}$$

Since in our experiments the cost function $c$ is binary and is a function of the state $c : \mathbb{S} \to \{0, 1\}$, the cost model can be used to predict the cost values of states. However, we propose to consider the safety of subgoals as the probability of their safety. Optimizing such safety helps generate goals that are further away from the boundary of safe and unsafe states. For the implementation of the cost model, we utilize a neural network (MLP approximator) parameterized by $\theta_c$. We use Binary Cross Entropy Loss as the loss function for learning this MLP. So, the output value of Cost Model can be considered as the probability that the state is safe. We train the model in online manner using the same buffer that is employed for the World Model.

The cost model does not have access to the complete state of the environment (i.e., the locations of all obstacles). Therefore, it cannot accurately predict the safety of states that are located at long

distances. Since HRAC predicts subgoals $s_g$ at short distances from the agent's current position, our experiments demonstrate that the agent's observations (i.e., local information about obstacles) are sufficient to accurately assess the safety of the subgoal $s_g$ during training.

**High-level policy objective.** To ensure that high level policy generates a safe subgoal, a component $L_{safety^h}(s, s_g) = C_{M_{\theta_c}}(s_g, o_{obst}, o_{pos})$ was incorporated into HRAC loss function (6) with scaling factor $\beta_{safe}^h$ (safety hyperparameter):

$$L^h = -Q^h(s, s_g) + \beta_{adj}^h L_{adj}(s, s_g) - \beta_{safe}^h L_{safety^h}(s, s_g), \tag{9}$$

where $\beta$ factors are predefined hyperparameters that are chosen empirically.

### 3.3 IMAGINATION SAFETY WITH WORLD MODEL: LOW-LEVEL SAFETY

The cost model is used to evaluate subgoals generated by the high-level policy and to optimize them to be safe subgoals. However, while the agent attempts to achieve these safe subgoals, it may violate safety constraints (for example, a mobile robot accelerating excessively while trying to reach a subgoal). To address this issue, we employ a combination of the cost model and the world model (the left learning scheme in Figure 2).

Given a current subgoal for the agent, $s_g$, we check for safety in imagination by sequentially generating actions using the low-level policy $a \sim \pi_{\theta_l}^l(a|s, s_g)$ and then employing the world model to obtain the next state $s' = M_{\theta_m}(s, a)$, repeating this procedure $n$ times. Here, $n$ is the number of imagination steps that is chosen from discrete uniform distribution $n \sim U(\{1..k-1\})$ (the left part of Figure 2). The total safety for low-level controller is calculated using the cost and the world models as follows:

$$safety^l(s, s_g) = C_{M_{\theta_c}}(\phi(s_n), o_{obst_i}, o_{pos_i}) \tag{10}$$

The resulting value of $safety^l(s, s_g) \in [0, 1]$ is then utilized in the total actor loss:

$$L^l = -Q^l(s, a, s_g) - \beta_{safe}^l L_{safety^l}(s, s_g); \ L_{safety^l}(s, s_g) = safety^l(s, s_g), \tag{11}$$

here $\beta_{safe}^l$ is a hyperparameter similar to $\beta_{safe}^h$ from (9). Exact values for used hyperparameters are presented in Table 4. Imaginary safety can be added to the high-level strategy as an additional loss rather than to the low-level one. However, since predicting subgoals is a more complex task, and based on our experiments, the low-level policy learns to achieve subgoals more quickly than the high-level policy, we incorporate imaginary safety into the low-level policy. To train the cost model and the world model, we utilize a warm start by executing $30,000$ random steps in the environment and pretrain both models over 100 epochs.

## 4 EXPERIMENTS

### 4.1 ENVIRONMENTS DESCRIPTION

**Long-Horizon environments.** SafeAntMaze depicted in Figure 3 was created with the safety wrapper for the MujocoAntMaze environment used in the work Zhang et al. (2020a). In SafeAntMaze, the agent is an Ant with action and observation spaces: $\mathbb{A} \subset \mathbb{R}^8, \mathbb{S} \subset \mathbb{R}^{30}$. The agent can only observe its current coordinates, joint angles, and angular velocities, lacking information about obstacles. Additionally, we have implemented a safety buffer for the agent's position: if it is within a specified distance, $dist$, from a wall, that position is considered unsafe, resulting in a cost of +1 for the agent. Since, on average, the optimal policy in this environment requires $\geq 500$ steps to reach the goal, the environment is classified as one that presents a long-horizon task. We developed two types of maps in this environment: SafeAntMazeCshape and SafeAntMazeWshape.

**Short-Horizon environments.** In the tasks from the Safety Gym benchmark (Ray et al., 2019), the optimal policy achieves the goal within 100 to 200 steps on average (see Figure 8). Therefore, we characterize this environment as short horizon. By default, the environment utilizes the Euclidean distance as the reward function. Additionally, we consider the PointGoal1 environment from SafetyGym, with sparse reward structure, granting a reward of +1 for reaching the goal. We refer to

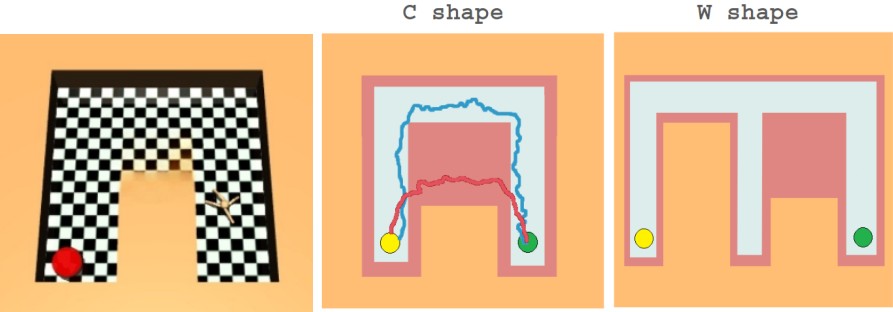

Figure 3: **Up** The figure illustrates the SafeAntMazeCshape environment, where a robot Ant is assigned, and the agent faces a long-horizon task with a specified goal. **Down:** The scheme of the environments for C shape(left) and W shape(right), where the *green point* represents Ant start pose, *yellow point* is Ant final goal, the *light field* represents a safe zone, the *pink field* denotes a dangerous area that incurs a cost of $+1$ for each step taken inside it, and the *orange zone* indicates a static obstacle. *The red trajectory* is generated by the HRAC algorithm, which does not take safety into account, while the *blue trajectory* is produced by the ITES algorithm.

this environment as PointGoal sparse. Training in an environment with sparse rewards is generally considered to be a more challenging task.

## 4.2 METRICS

Following Ray et al. (2019), one method dominates another if it strictly improves on either performance or cost rate and does at least as well on the other.

- Performance — average success rate(reward) of the final policy over $E = 40$ episodes
- Average undiscounted cost return over E episodes: $\hat{J}_c = \frac{1}{E} \sum_{i=1}^{E} \sum_{t=0}^{T_{ep}} c_t$
- Average cost rate: $\rho = \frac{1}{T} \sum_{t=0}^{t=T} c_t$, where $T$ - total interaction time steps

## 4.3 BASELINES COMPARTION

We consider two types of baselines: hierarchical (HRAC, HRAC-LAG) and those consisting of a single policy: CUP (Yang et al., 2022), FOCOPS (Zhang et al., 2020b), MBPPOL (Jayant & Bhatnagar, 2022) TD3LAG Ray et al. (2019). In our approach, we utilize the environment model solely for calculating the safety for the low-level policy; we do not use it for generating additional experience or planning. The training is conducted using a model-free approach; therefore, we compare ITES with model-free baselines.

- **HRAC - LAG**. To obtain a fair comparison of our safety component to HRAC algorithm against the Lagrangian-based approach, we added a Lagrange multiplier to the high-level policy of the HRAC algorithm and named this as HRAC - LAG. This way, similar performance is expected from both HRAC-LAG and ITES.
- **HRAC -SafeSubgoals**. To investigate the impact of safety incorporated solely within high-level policy, we remove the safety component from the low-level policy, resulting in HRAC - SafeSubgoals
- **HRAC - SafeController**. To assess how safety verification in imagination affects cost return, we remove the generation of safe subgoals from ITES, resulting in HRAC - Safe-Controller.
- **MBPPOL**. To compare with model-based approaches, we propose the MBPPOL algorithm, an algorithm consisting of a single policy that utilizes a world model to train a Lagrangian PPO in a simulated environment.
- **TD3LAG**. To provide a comparison with off-policy approaches, we propose the TD3LAG algorithm, which consists of the TD3 policy and a Lagrangian multiplier.

Table 1: **Final performance on SafeAntMaze.**

| Env | Method | Final Success Rate | Final Cost |
|---|---|---|---|
| SafeAntMaze Cshape | ITES | $\mathbf{0.88} \pm 0.01$ | $15.2 \pm 17.3$ |
| | CUP | $0$ | $\mathbf{6.8} \pm 22.4$ |
| | FOCOPS | $0$ | $19.4 \pm 50.9$ |
| | TD3LAG | $0.38 \pm 0.04$ | $83.6 \pm 37.2$ |
| SafeAntMaze Wshape | ITES | $\mathbf{0.35} \pm 0.05$ | $183 \pm 16.5$ |
| | TD3LAG | $0.05 \pm 0.02$ | $\mathbf{154} \pm 13.5$ |

In Figure 4, we present a comparison on SafeAntMaze environments of our proposed approach, ITES, with the HRAC-LAG algorithm, which implements safety through a Lagrangian multiplier, as well as with the HRAC algorithm that does not account for safety. The success rate plot indicate that each approach achieves a similar performance of the final policy, ranging from 0.89 to 0.93, suggesting comparable performance. However, in terms of cost and cost rate metrics, our algorithm significantly outperforms the Lagrangian-based HRAC. This discrepancy arises because optimizing the safety component within the high-level policy of HRAC-LAG is considerably more complex than in our proposed method. In HRAC-LAG, the safety critic attempts to approximate the cumulative safe cost that the agent accumulates while reaching a subgoal, which is contingent upon the current behavior of the low-level policy. In contrast, in our approach, the safety of the high-level module is independent of the behavior of the low-level policy.

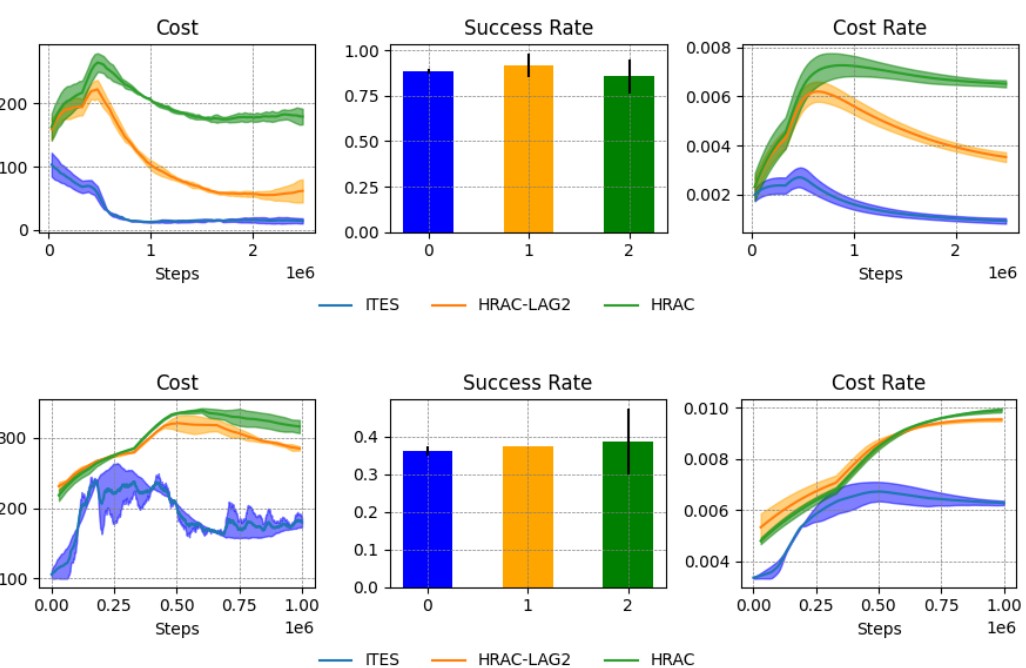

Figure 4: **Baselines comparison on SafeAntMaze**. Comparison of our proposed ITES method with the safe HRAC-LAG method and the unsafe HRAC method on SafetyAntMaze. First row — SafeAntMazeCshape environment, second row — the SafeAntMazeWshape environment. Each run was conducted with 3 seeds. The shaded area represents the standard deviation. The Success Rate was calculated based on the weights at the end of the training.

In the experiments on the SafetyGym benchmark, Figure 5 presents a comparison with hierarchical policies. It is observed that in the CarGoal1 environment, our method outperforms HRAC-LAG in

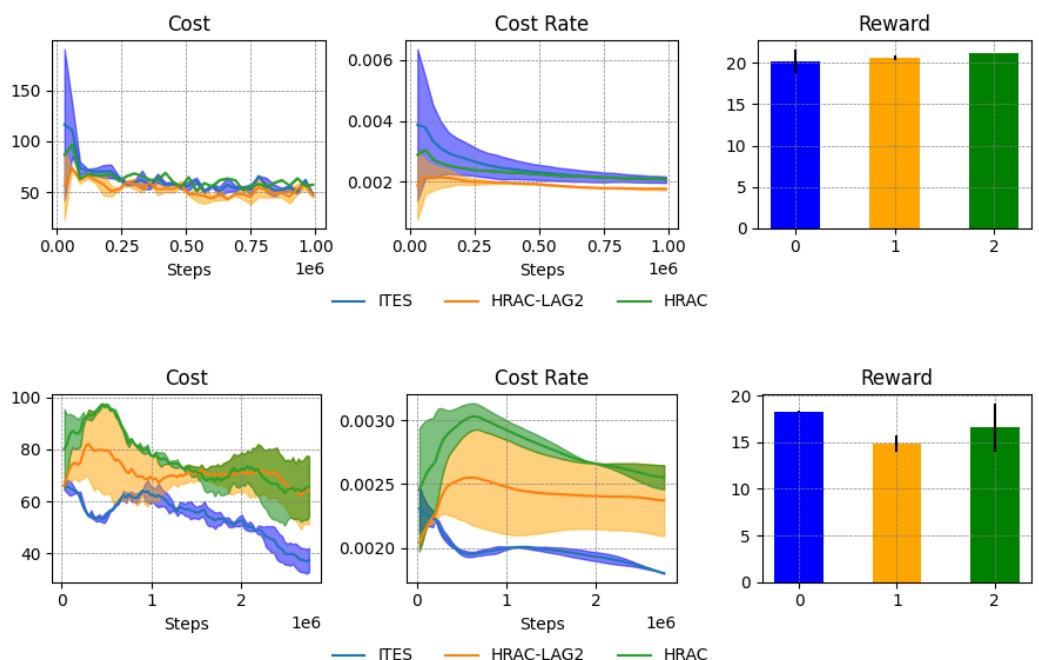

Figure 5: **Baselines comparison on SafetyGym**. First row - PointGoal1, second row - CarGoal1

Table 2: **Final performance on SafetyGym benchmark.**

| Env | Method | Final Reward | Final Cost |
|---|---|---|---|
| PointGoal1 | ITES | $\mathbf{21.42} \pm 1.14$ | $40.46 \pm 0.12$ |
| | CUP | $14.42 \pm 6.74$ | $\mathbf{19.02} \pm 20.08$ |
| | FOCOPS | $14.97 \pm 9.01$ | $33.72 \pm 42.24$ |
| | TD3LAG | $15.75 \pm 0.39$ | $56.22 \pm 0.85$ |
| CarGoal1 | ITES | $\mathbf{19.43} \pm 3.45$ | $39.87 \pm 1.04$ |
| | CUP | $6.14 \pm 6.97$ | $36.12 \pm 89.56$ |
| | FOCOPS | $15.23 \pm 10.76$ | $\mathbf{31.66} \pm 93.51$ |
| | TD3LAG | $8.21 \pm 6.88$ | $53.7 \pm 9.45$ |
| PointGoal sparse | ITES | $\mathbf{8.64} \pm 0.38$ | $\mathbf{33.46} \pm 1.47$ |
| | MBPPOL | $6.15 \pm 0.15$ | $\mathbf{34.1} \pm 3.6$ |
| | SACLAG | $0.68 \pm 0.08$ | $64.8 \pm 3.8$ |
| | TD3LAG | $0.07 \pm 0.01$ | $64.7 \pm 1.91$ |

both safety and performance. In contrast, in the PointGoal1 environment, all three approaches are nearly identical.

Table 2 provides a comparison of CUP and FOCOPS final weights on SafetyGym. For the Point-Goal1 task, our approach lags behind CUP in terms of safety, as it makes a trade-off in favor of performance and solves significantly more tasks, achieving a reward of 21.42. Moreover, it is a less dispersion method, indicating its reliability. In the CarGoal1 task, our method has a comparable final cost while significantly outperforming the CUP algorithm in terms of performance. The TD3-Lag algorithm demonstrates superior performance compared to CUP and FOCOPS; however, this results in a significant increase in the final cost metric, whereas ITES remains safer while improving performance.

The results on the GoalPoint sparse environment (see Table 2 ) indicate that simple model-free single policy algorithms are unable to learn to solve the task, with performance scores of SAC-L at 0.675

and TD3-L at 0.07. For the results presented in the table, MBPPOL used weights that achieved the highest performance during training, while ITES utilized weights with a similar cost metric during training. These results demonstrate that, under similar safety conditions, ITES exhibits superior performance compared to MBPPOL, achieving a score of 8.64.

### 4.4 INFLUENCE OF EACH SAFETY LEVEL ON OVERALL SAFETY

To address the question - "Can safety be managed solely at one level, such as the low-level controller?" - we conducted a comparative study of the ITES algorithm against HRAC-SafeSubgoals (which enforces safety only at the high-level controller) and HRAC-SafeController (which enforces safety solely at the low-level controller) within the SafeAntMazeCshape environment. As depicted in Figure 6, the cost and cost rate metrics indicate that in the SafeAntMazeCshape environment, the high-level controller significantly enhances safety. Nonetheless, we observe that incorporating low-level safety alongside high-level safety yields an overall increase in safety. Additionally, it is noteworthy that HRAC-SafeSubgoals exhibits greater variance in the cost plot compared to ITES, which results from the fact that when the HRAC-SafeSubgoals policy reaches a safe subgoal, it may violate safety along the way.

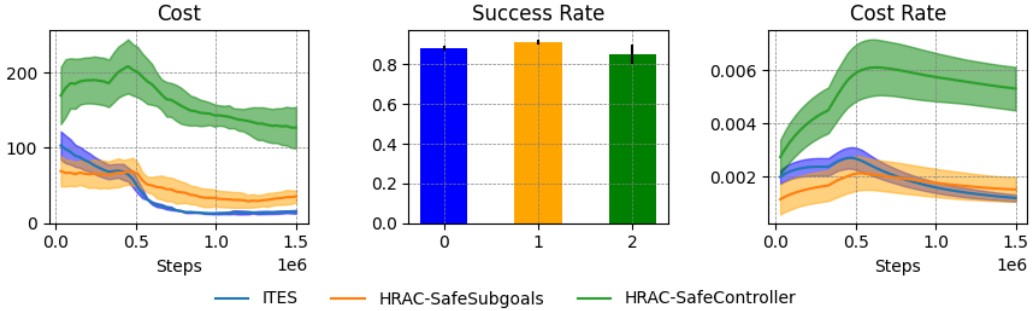

Figure 6: **Low/High safety level analysis**. Different versions of ITES are depicted in the figure, with HRAC-SafeSubgoals implementing safety solely at the high level and HRAC-SafeController implementing safety solely at the low level.

## 5 RELATED WORKS

Safe reinforcement learning aims to develop algorithms that ensure safety both during the learning process (exploration phase) and the exploitation phase within an environment. A widely adopted approach to guarantee safety in RL tasks involves solving a Constrained Markov Decision Process using the Lagrange Multiplier method (Ray et al., 2019). This approach addresses two key objectives: maximizing cumulative reward while minimizing cumulative cost, effectively balancing performance and safety.

While traditional reinforcement learning algorithms such as SAC (Haarnoja et al., 2018), DDPG (Lillicrap et al., 2016), and Dreamer (Hafner et al., 2023) are primarily engineered to optimize cumulative reward, the integration of the Lagrange Multiplier method transforms this paradigm. By converting the dual CMDP objective into a single one, the method introduces a trade-off into the value function space, thus enables balancing between performance and safety constraints. Algorithms like SAC-Lagrangian (Ha et al., 2021), SafeSLAC (Hogewind et al., 2023), SafeDreamer (Huang et al., 2024), NeuralConditionedSAC (Huang et al., 2021) embodied the Lagrange approach within their architectures. During updates, the policy takes into account the sum of the values from the reward and cost critics, there the Lagrange multiplier scales the cost component.

Despite of the widespread use of the method and its strong sides, it suffers from instabilities and oscillations during learning due to the non-stationary nature of the dual objectives during policy optimization. The effective solution to this challenge was proposed by Stooke et al. (2020). The idea was taken from control theory: it was proposed to use well-known PID (Proportional-Integral-

Derivative) controller to update a Lagrange multiplier, whereas the classical approach uses only the Integral part.

Another way to mitigate problems of the Lagrangian method, an alternative approach, is to introduce additional layers into the policy architecture, dedicated solely to ensuring safety by correcting actions as necessary. Such algorithms are presented by Safe HIRO (Roza et al., 2022), SafetyLayer (Dalal et al., 2018), SafeEditor (Yu et al., 2022). This decoupling of the safety mechanism from the reward-cost value function aggregation offers a promising direction to reduce stochasticity while maintaining the robustness and reliability of the agent's behavior.

Serious challenge in safe exploration is the unknown dynamics of the environment. Without any prior information the agent is forced to violate some constraints just to know about them. If an agent has been equipped with a world model it can learn safe policy with much less safety violations as using the world model "imagination" is safe in contrast to interaction with real world. With image observations this approach is presented by (Huang et al., 2024), the SOTA model-based algorithm adopted PIDLagrangian (Stooke et al., 2020). Also there exist several works (Hogewind et al., 2023; Liu et al., 2020; Jayant & Bhatnagar, 2022). (Huang et al., 2024) used not only images as observations and also vector states.

To improve exploration in RL it is typically used hierarchy approaches, that allows to reduce complexity of the task by decomposing it into subtasks. These approaches involve generating high-level actions, subgoals, that the policy must achieve within a limited number of low-level actions: HAC (Levy et al., 2019), HiRO (Nachum et al., 2018), HRAC (Zhang et al., 2020a).

In Safe RL, very few works consider hierarchical approaches. Among them the most closed to us is SafetyLayer+HiRO (Roza et al., 2022). In contrast to us it solves additional optimization task for each action to replace it with a safe one, whereas our algorithm is optimized only during training phase.

To the best of our knowledge, our algorithm ITES is the only one that combines model-based and hierarchy approach with safety constrained optimization.

## 6 CONCLUSIONS

In summary, we investigated the Safe Exploration Problem, where the agent must maximize performance while minimizing safety violations. We proposed the ITES approach based on the HRAC algorithm, which provides safety at two levels of the hierarchy. Additionally, we explored an alternative implementation of safety in HRAC using a Lagrangian multiplier, referred to as HRAC-LAG. Experimental results demonstrated that ITES achieves comparable performance to HRAC-LAG while significantly enhancing safety during training (safe exploration). In the SafetyGym environment, we demonstrated that our algorithm achieves significantly higher performance while maintaining comparative safety and exhibiting lower dispersion.

However, ITES has limitations, including the need to manually design the mapping function $\phi$ from state space to goal space for each task, which restricts its adaptability, particularly to visual input. Additionally, the simple discretization technique used by HRAC's Adjacency Network limits its scalability to high-dimensional spaces. Furthermore, ITES does not account for task-specific cost budgets $d$, as it minimizes cost violations independently at each time step.

Future work will focus on addressing these limitations, such as learning goal spaces, improving scalability, and incorporating cost budgets, to enable the application of ITES in real-world robotics tasks.

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

## A COMPARISON OF MODEL-BASED SAFETY AND HIERARCHICAL + MODEL-BASED SAFETY

The results of training ITES and TD3 with safety, which we calculate in ITES using the environment model (TD3-imagine safety), are presented in Figure 7 and Table 3. Although both algorithms show similar values in the Reward metric, given that the experiment is conducted in the PointGoal1 environment, which is not a long - horizon task, the cost metric indicates that ITES is significantly safer than TD3-imagine safety. This difference arises because TD3-imagine safety can only minimize safety in imagination for 10 steps ahead (not until the end of the episode); we cannot increase this number, as doing so would cause the world model to diverge. In contrast, ITES generates subgoals for 10 steps ahead, enabling the TD3 policy to safely reach these goals.

Table 3: **Final performance on PointGoal1** comparing Model-based and Hierarchical Model-based safety.

| Env | Method | Final Reward | Final Cost |
|---|---|---|---|
| | ITES | $21.42 \pm 1.14$ | $40.46 \pm 0.12$ |
| PointGoal1 | TD3-imagine safety | $20.1 \pm 0.1$ | $75 \pm 10.4$ |

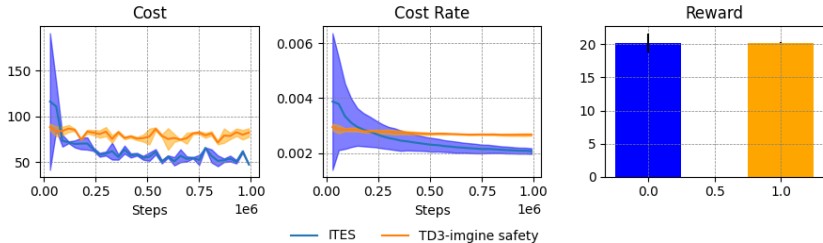

Figure 7: **Comparison of Model-Based Safety and Hierarchical + Model-Based Safety.** Plots were obtained in the PointGoal1 environment across three seeds.

## B SAFETYGYM ENVIRONMENT

## C TRAINING DETAILS

The training procedure for our ITES algorithm is based on HRAC (Zhang et al., 2020a), with the addition of two key components: the world model $M$ and the cost model $C_M$, which are integrated into the process (see Algorithm 1). Initially, $M$ and $C_M$ are pretrained using data generated by a random policy. During the main interaction loop, transitions are stored in buffers $\mathcal{B}, \mathcal{B}_h, \mathcal{B}_l$. After each episode of agent-environment interaction, all trainable components of ITES are updated. These components include the cost model $C_M$, world model $M$, high-level policy $\pi^h$, and low-level

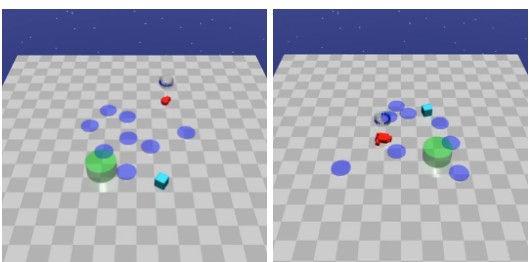

Figure 8: The image presents two tasks: PointGoal1 (left) and CarGoal1 (right) from the SafetyGym environment. *The blue circles* represent hazardous zones, where the agent incurs a cost of +1 while within them. *The green circle* indicates the agent's goal, and the *blue box* represents a movable vase, which the agent can interact with without incurring a penalty.

policy $\pi^l$. The adjacency network $\psi$ is updated every $L$ episodes, where $L$ corresponds to 50,000 environment steps.

The world model is utilized as described in Jayant & Bhatnagar (2022), and details of its training can be found in that work. The learning procedure for the adjacency network is outlined in Zhang et al. (2020a).

Loss functions for policy training, along with details on the cost model training, are presented in the main text of the article (Section 3).

A complete list of hyperparameters is provided in Table 4. Most of these were adopted from the corresponding algorithms HRAC (Zhang et al., 2020a) and MBPPOL (Jayant & Bhatnagar, 2022). For the Car and Point environments, the same hyperparameters were used (SafetyGym column). Similarly, the hyperparameters for the SafeAntMaze(C shape) and SafeAntMaze(W shape) environments are identical (SafeAntMaze column).

---

**Algorithm 1** ITES

---

**Input:** High-level policy $\pi^h$, low-level policy $\pi^l$, world model $M$, cost model $C_M$, adjacency network $\psi$, goal transition function $h$, high-level action frequency $k$, number of training episodes $N$, adjacency learning frequency $L$, empty adjacency matrix $\mathcal{M}$, empty trajectory buffer $\mathcal{B}_\mathcal{A}$, cost and world model buffer $\mathcal{B}$, empty high-level and low-level policy buffers $\mathcal{B}_h, \mathcal{B}_l$.

Sample and store trajectories in the model buffer $\mathcal{B}$ using a random policy.
Pretrain $M$ and $C_M$ using $\mathcal{B}$.
**for** $n = 1$ **to** $N$ **do**
   Reset the environment and sample the initial state $s_0$.
   $t = 0$.
   **repeat**
     **if** $t \equiv 0 \,(\mathrm{mod}\, k)$ **then**
       **if** $t \neq 0$ **then**
         Save subgoal transition $\langle s_{t-k}, s_{g,t-k}, s_t, \sum_{i=t-k}^{t-1} r_i, done_t \rangle$ to buffer $\mathcal{B}_h$
       **end if**
       Sample subgoal $s_{g,t} \sim \pi^h_(s_{g,t}|s_t)$.
     **else**
       Perform subgoal transition $s_{g,t} = h(s_{g,t-1},\ s_{t-1},\ s_t)$.
     **end if**
     Sample low-level action $a_t \sim \pi^l_{\theta_l}(a_t|s_t,\ s_{g,t})$.
     Make action in environment:
       Sample next state $s_{t+1} \sim p(s_{t+1}|s_t,\ a_t)$,
       Get reward $r_t = r(s_t, a_t, s_{t+1})$,
       Get cost $c_t = c(s_t, a_t)$,
       Get episode end signal $done_{t+1}$.
     Calculate reward $r_{in,t} = r_{in}(s_{t+1}, s_{g,t})$,
     Save transition $\langle s_t, a_t, r_{in,t}, c_t, s_{t+1}, done_{t+1} \rangle$ to buffers $\mathcal{B}, \mathcal{B}_l$
     $t = t + 1$.
   **until** $done_{t+1}$ is $true$.
   Store the sampled trajectory in $\mathcal{B}_\mathcal{A}$.
   Train cost model $C_M$.
   Train world model $M$.
   Train high-level policy $\pi^h$.
   Train low-level policy $\pi^l$.
   **if** $n \equiv 0 \,(\mathrm{mod}\, L)$ **then**
     Update the adjacency matrix $\mathcal{M}$ using the trajectory buffer $\mathcal{B}_\mathcal{A}$.
     Fine-tune $\psi$ using $\mathcal{M}$.
     Clear $\mathcal{B}_\mathcal{A}$.
   **end if**
**end for**

---

Table 4: ITES hyperparameters.

| Hyperparameter | SafeAntMaze | SafetyGym |
|---|---|---|
| **Adjacency Network Parameters** | | |
| Learning rate | 0.0002 | 0.0002 |
| Batch size | 64 | 64 |
| Online training frequency (steps) | 50, 000 | 50, 000 |
| Online training epochs | 25 | 25 |
| Embedding dim | 32 | 32 |
| Hidden dim | 128 | 128 |
| $\epsilon_k$ | 1.0 | 1.0 |
| $\delta$ | 0.2 | 0.2 |
| | | |
| **Manager Parameters (High-level TD3)** | | |
| Actor learning rate | 0.0001 | 0.0001 |
| Critic learning rate | 0.001 | 0.001 |
| Replay buffer size | 200, 000 | 200, 000 |
| Batch size | 128 | 128 |
| Soft update rate | 0.005 | 0.005 |
| Policy update frequency (steps between updates) | **10** | **5** |
| $\gamma$ | 0.99 | 0.99 |
| High-level action frequency $k$ | **20** | **10** |
| Reward scaling | **0.1** | **100** |
| Adjacency loss coefficient $\beta_{adj}^h$ | 20 | 20 |
| | | |
| **Controller Parameters (Low-level TD3)** | | |
| Actor learning rate | 0.0001 | 0.0001 |
| Critic learning rate | 0.001 | 0.001 |
| Replay buffer size | 200, 000 | 200, 000 |
| Batch size | 128 | 128 |
| Soft update rate | 0.005 | 0.005 |
| Policy update frequency | 1 | 1 |
| $\gamma$ | 0.95 | 0.95 |
| | | |
| **Cost Model Parameters** | | |
| Initial exploration steps | **10, 000** | **30, 000** |
| Pretrain epochs | **20** | **100** |
| Batch size | **128** | **512** |
| Buffer size | 1, 000, 000 | 1, 000, 000 |
| Learning rate | 0.001 | 0.001 |
| | | |
| **WorldModel Parameters** | | |
| Initial exploration steps | **10, 000** | **30, 000** |
| Pretrain epochs | **20** | **100** |
| Batch size | 256 | 256 |
| Buffer size | 1, 000, 000 | 1, 000, 000 |
| Learning rate | 0.001 | 0.001 |
| Train freq | 20 | 20 |
| Num networks | 8 | 8 |
| Num elites | 6 | 6 |
| Hidden size | 200 | 200 |
| | | |
| **Safety Parameters** | | |
| Subgoal safety coefficient $\beta_{safe}^h$ | **800** | **10** |
| Controller safety coefficient $\beta_{safe}^l$ | **6** | **0.001** |

