# OpenReview forum: "Imagine to Ensure Safety in Hierarchical Reinforcement Learning"
_ICLR.cc/2025/Conference — Submitted to ICLR 2025_

### Official Review · Reviewer_nj6z · 2024-10-19

**Soundness:** 3
**Presentation:** 2
**Contribution:** 2
**Rating:** 5
**Confidence:** 5

**Summary:**

The core innovation of this paper is the proposed method called ITES (Imagine To Ensure Safety in Hierarchical Reinforcement Learning). This method combines model-driven reinforcement learning and hierarchical reinforcement learning to ensure safety. It introduces a high-level policy and a low-level policy to generate safe intermediate subgoals and guide the agent toward the final goal. Through trajectory imagination, the low-level policy learns to reach these subgoals safely.
Key contributions include:
Using hierarchical reinforcement learning for enhanced safety and performance.
Employing a world model to verify safety in imagination and a cost model for generating safe subgoals.
Demonstrating superior performance on SafetyGym and SafeAntMaze benchmarks compared to state-of-the-art methods, especially in satisfying safety constraints.

**Strengths:**

Compared to existing methods, ITES (Imagine To Ensure Safety in Hierarchical Reinforcement Learning) achieves better performance quality while maintaining comparable safety violations. Additionally, the paper explores an alternative implementation of safety in hierarchical reinforcement learning using Lagrange multipliers and demonstrates that ITES provides better safety while maintaining comparable performance to traditional safe policies in a custom long-horizon environment called SafeAntMaze.

**Weaknesses:**

1.The paper claims novelty in integrating safety with hierarchical reinforcement learning (HRL) and a model-based approach, but similar methods have been explored previously. For instance, Safe HIRO [1]  and IAHRL [2] also focuses on hierarchical approaches for safe exploration.  Provide a clearer distinction between ITES and existing approaches. Explicitly state the unique aspects of ITES, such as how the world model or subgoal generation differs from previous methods.
[1] Roza F S, Roscher K, Günnemann S. Safe and efficient operation with constrained hierarchical reinforcement learning[C]//Sixteenth European Workshop on Reinforcement Learning, 2023.
[2] Lee S H, Jung Y, Seo S W. Imagination-augmented hierarchical reinforcement learning for safe and interactive autonomous driving in urban environments[J]. IEEE Transactions on Intelligent Transportation Systems, 2024.
2.The use of a world model for safety verification in imagination depends on the accuracy of the learned model. If the model is imperfect, it may not accurately detect safety violations, potentially leading to unsafe behavior in the real environment. This paper does not address the limitations of model inaccuracies or present any strategies to handle model uncertainty.

**Questions:**

1.How exactly is the safe region for subgoal generation defined, and how is it updated during the training process? Is it a static boundary, or does it adapt based on the agent's learning progress?
2.Provide a clearer distinction between ITES and existing approaches, such as [1] and [2].
[1] Roza F S, Roscher K, Günnemann S. Safe and efficient operation with constrained hierarchical reinforcement learning[C]//Sixteenth European Workshop on Reinforcement Learning, 2023.
[2] Lee S H, Jung Y, Seo S W. Imagination-augmented hierarchical reinforcement learning for safe and interactive autonomous driving in urban environments[J]. IEEE Transactions on Intelligent Transportation Systems, 2024.
3.Have you considered any potential challenges in transferring the proposed ITES method to real-world robotic applications? How would the approach handle noisy sensor data or unexpected changes in the environment?

---

> ### Author Response · Authors · 2024-11-25
>
> Thank you for valuable questions and commentaries. We provide detailed answers for them below.
>
> # Weaknesses:
>
> - Comparison ITES with Safe HIRO and IAHRL.
>
> Comparison ITES and Safe HIRO:
>
> The Safe HIRO algorithm consists of two modules: a low-level learnable policy and a high-level HIRO policy. To ensure safety, Safe HIRO employs a Safety Layer, which substitutes each greedy action of the low-level policy with a safe action. The first distinction between Safe HIRO and ITES is that Safe HIRO only focuses on the safety of the controller's actions, without considering the safety of the high-level goal, which can result in the agent entering unsafe areas. In contrast, ITES addresses safety concerns at both levels, ensuring that we always remain in safe regions. The second distinction is that ITES utilizes a world model, which prevents the agent from reaching dangerous states and allows for the maximization of safety in imagination. This approach enhances the sample efficiency of the algorithm compared to Safe HIRO.
>
>
> Comparison ITES and IAHRL:
>
> The IAHRL algorithm consists of a high-level policy (learned via the RL algorithm SAC) and a set of low-level controllers (each of which is a predefined Frenet planner). In contrast, ITES employs a single learnable low-level policy that is conditioned on arbitrary goals within the environment.The imagination-based safety in ITES relies on a learnable world model, which is continuously updated with newly acquired experience and is applicable to any task. In IAHRL, safety is hardcoded into the planner, which is specifically designed for autonomous driving tasks In contrast, ITES does not have such limitations and can be utilized across a variety of environments. Additionally, subgoal safety in ITES is defined using a learnable Cost Model, which incorporates a safety component directly into the loss function of the high-level policy. In IAHRL, safety is implemented as a component of the reward, this leads to the well-known problem of reward engineering. While ITES addresses a more general formulation of CMDP, where the issue of reward engineering does not arise.
>
> - We completely agree that the quality and confidence of the world model are crucial for safety-critical tasks. We recognized this from the outset and designed ITES with this consideration in mind. To minimize model uncertainty in ITES, we employ a classic strategy for reducing world model errors—learning an ensemble of world models. Additionally, our cost model is trained as a probabilistic classifier, providing the probability that a given state is safe or unsafe.
>
>
> # Questions:
> - The safe region is implicitly defined by our learnable cost model: ITES learns this cost model from data generated through interactions with the environment. As the cost model is updated, the agent's understanding of the safe region is also refined. We do not use any fixed threshold to distinguish between two classes of states. Instead, ITES aims to minimize this probability to zero as part of the optimization process (we acknowledge in the conclusion section that this is one of ITES's limitations).
> - The distinction between ITES and Safe HIRO, as well as IAHRL, is clearly outlined above (in the first point of the responses to the weaknesses).
> - We consider the transfer of proposed ITES to real-world robotic applications as our future research. About unexpected changes: as our proposed algorithm is learnable it will adapt to changes in an environment, however to make possible real-world applications we should take into account that during the adaptation the agent policies become more unsafe.

---

> > ### Author Response · Authors · 2024-11-27
> >
> > We have made revisions based on the recommendations of all reviewers. We would like to invite you to review our current version of the manuscript. Do you have any preferences regarding further improvements to our work?

---

> > ### Comment · Reviewer_nj6z · 2024-11-28
> > **Thanks for your response!**
> >
> > Thank you for your positive reply, which has solved most of my questions. However, the reply from Comparison ITES with Safe HIRO and IAHRL did not convince me completely. Can you carry out specific experimental effects under the same conditions?

---

> > > ### Author Response · Authors · 2024-12-02
> > >
> > > Thank you for your constructive suggestions.
> > >
> > > Regarding IAHRL, we want to emphasize that the IAHRL algorithm is **specifically** designed to solve problems related **to autonomous driving**, which significantly **limits** the potential application of the algorithm **to a certain type of tasks**. Tasks from IAHRL paper are not publicly available, nor is the implementation of this algorithm, making it impossible to conduct experimental comparisons with this method in the short term.
> > >
> > > For the SafeHIRO algorithm, the implementation of the agent is also not in open access; however, the tasks discussed in the paper are accessible, allowing us to conduct such a comparison with this algorithm.
> > >
> > > We conducted experiments comparing SafeHIRO and ITES on the Bullet Safety Gym benchmark. The results for SafeHIRO were obtained from a study after 700K steps, and ITES was also trained for 700K steps.
> > >
> > > | Env | Method | Final Reward | Final Cost |
> > > | ------------------- | ------------- | ---------------- | ---------------- |
> > > | SafeCarRun sparse | ITES | $347$| $ -217 $ |
> > > | SafeCarRun sparse | SafeHIRO | $ - $ | $ - $ |
> > > | SafeCarRun dense | ITES | $347$ | $-217$ |
> > > | SafeCarRun dense | SafeHIRO | $ 216 $ | $ -480 $ |
> > >
> > >
> > > CurRun environment to a dense value (adding information about the distance to walls into the cost); this environment is indicated in the table as 'cost dense.' Since the SafeLayer used in SafeHIRO requires a differentiable cost function for training, SafeHIRO cannot be trained in the original formulation of the problem in an environment where the cost is a binary sparse function, which is why for SafeHIRO the dashes are in the 'sparse' row.
> > >
> > > ITES can be trained on both sparse and dense cost functions and yields the same results without entering the dangerous zone (where sparse cost = 0). From the results obtained with sparse cost, it is evident that the cost value for SafeHIRO is lower, as they utilize additional information in the cost function—specifically, the distance to obstacles in the form of a cost signal—while our algorithm does not use such information. However, our algorithm outperforms SafeHIRO in terms of performance.
> > >
> > > | Method | General adaptive low-level policies | Doesn't require cost function differentiability | Safety of subgoals | World Model |
> > > | ------------------- | ------------------- | ------------------- |  ------------------- | ------------------- |
> > > |  ITES| + |  + | +  |  +  |
> > > |  SafeHIRO | + |  -  |  -  | - |
> > > | IAHRL | - |  +  | + |  +  |

---

> > > > ### Comment · Reviewer_nj6z · 2024-12-02
> > > > **Thank you for your positive reply！**
> > > >
> > > > Thank you for your positive reply！

---

### Official Review · Reviewer_6mim · 2024-11-01

**Soundness:** 2
**Presentation:** 3
**Contribution:** 2
**Rating:** 6
**Confidence:** 3

**Summary:**

The paper proposes ITES (Imagine To Ensure Safety), a hierarchical reinforcement learning (HRL) method that incorporates a world model and dual-level safety policy to address the safe exploration problem. By generating safe subgoals with a high-level policy and optimizing short-horizon safety at the low level, ITES aims to balance performance and safety effectively. The world model enables ITES to "imagine" potential actions, verifying their safety before executing them in the real environment. Experiments on SafetyGym tasks and a more complex SafeAntMaze environment suggest that ITES achieves a safety advantage in long-horizon tasks while maintaining competitive performance in simpler settings.

**Strengths:**

ITES’s use of dual-level safety with high-level safe subgoal generation and low-level safety checks is a well-conceived approach to enhancing reinforcement learning safety. The method of verifying actions in imagination through a world model adds an innovative layer that may reduce real-world safety violations.

The results show that ITES achieves a reasonable balance between safety and performance, particularly in SafeAntMaze. This structure helps demonstrate ITES’s suitability for scenarios where long-horizon safety is critical, even if it compromises some performance in short-horizon tasks.

ITES introduces a practical approach for managing safety constraints in HRL by embedding safety checks at both the planning and execution levels, which may inspire further hierarchical developments in safe reinforcement learning.

**Weaknesses:**

Although ITES demonstrates safety advantages in the complex SafeAntMaze environment, it does not consistently provide a safer solution than CUP in simpler SafetyGym tasks. For example, in the PointGoal1 task, ITES sacrifices some safety in favor of performance, resulting in a higher reward but slightly increased safety violations compared to CUP. This suggests that ITES may not consistently prioritize safety over performance across different task types, which could limit its applicability in certain safety-critical environments.

SafeAntMaze is the only complex, long-horizon task tested in the paper, which restricts the evidence supporting ITES’s generalizability to other challenging environments. Without more complex benchmarks, it’s difficult to conclude that ITES is superior to other approaches in handling diverse safety-intensive scenarios.

In the short-horizon SafetyGym tasks, ITES shows a performance advantage over CUP in the CarGoal1 task but lacks consistent improvements in safety. This performance-safety trade-off indicates that ITES may be best suited to tasks where performance can be favored without compromising critical safety constraints, making it less optimal for environments that strictly require safety prioritization.

The hierarchical structure in ITES introduces added complexity, but the benefits of this structure are not consistently clear across all tasks. While ITES demonstrates advantages in SafeAntMaze, the mixed results in SafetyGym suggest that simpler, single-policy methods may sometimes offer comparable or superior performance and safety. A broader range of tests could provide a clearer justification for the added complexity of ITES’s hierarchical approach.

**Questions:**

Could the authors provide further evidence or analysis supporting ITES’s necessity in complex environments by testing on additional long-horizon tasks?

Given that ITES sacrifices some performance in short-horizon tasks, what are the potential trade-offs in settings where safety constraints are less critical?

Could the authors clarify if ITES could be combined with other model-free or model-based approaches to mitigate the observed performance gaps in simpler environments?

---

> ### Author Response · Authors · 2024-11-25
>
> Thank you for your valuable feedback and constructive suggestions.
> # Weaknesses:
> - You are correct that in the PointGoal1 environment, our method exhibits a greater trade-off towards performance rather than safety. We propose that in this environment, it is impossible to optimize performance without sacrificing safety. To support this hypothesis, we investigated an additional baseline TD3LAG which demonstrates higher performance (15.75) compared to CUP (reward = 6.14 ± 6.97) and FOCOPS (reward = 5.23 ± 10.76), albeit at a significant cost to safety (cost = 56.22 ± 0.85) when compared to our method ITES (cost = 40.46 ± 0.12). Although our method generally exhibits lower safety than two out of three baselines in the PointGoal1 environment, it is more reliable in terms of variance: CUP (cost_std = 89.56), FOCOPS (cost_std = 93.51), TD3LAG (cost_std = 9.45), ITES (cost_std = 1.04). The substantial variance observed in CUP and FOCOPS indicates that there are seeds (random conditions) under which they violate safety constraints excessively, whereas our algorithm displays a more stable safety. To ensure that our method provides a favorable trade-off towards safety in more safety-critical environments, we recommend utilizing hyperparameters that address safety specifically, the “Subgoal Safety Coefficient” and “Controller Safety Coefficient”, as outlined in Appendix Table 4.
> | Env | Method | Final Reward | Final Cost |
> | ------------------- | ------------- | ---------------- | ---------------- |
> |   PointGoal1| ITES | $21.42 \pm 1.14$| $40.46 \pm 0.12$ |
> | PointGoal1 | CUP | $14.42 \pm 6.74$ | $19.02 \pm 20.08$ |
> | PointGoal1 | FOCOPS | $14.97 \pm 9.01$ | $33.72 \pm 42.24$ |
> | PointGoal1 | TD3LAG | $ 15.75 \pm 0.39$ | $56.22 \pm 0.85$ |
>
>
> - Thank you for your suggestion to improve the generalizability of our algorithm. To expand the range of complex environments, we have added SafeAntMaze W shape, an additional map for the SafeAntMaze environment that requires more steps to solve the task, and we have tested it with (ITES, HRAC, HRACLAG, TD3LAG). We have also introduced the SafetyGymPointGoal sparse environment, which consists of the same tasks as PointGoal1, but where a reward of +1 is granted only for reaching the goal, significantly increasing the difficulty of learning. In both of the added environments, model-free single-policy methods are unable to solve the tasks, highlighting the need for a hierarchical approach.
> | Env | Method | Final Reward | Final Cost |
> | ------------------- | ------------- | ---------------- | ---------------- |
> |   PointGoal  sparse  | ITES | $8.64 &pm; 0.38$| $33.455 &pm; 1.47$ |
> | PointGoal  sparse | MBPPOL | $6.15 &pm; 0.15$ | $34.1 &pm; 3.6$ |
> | PointGoal  sparse | SAC-L| $0.675 &pm;0.08$ | $64.8 &pm; 3.8$ |
> | PointGoal  sparse | TD3-L | $ 0.07 &pm; 0.006$ | $64.7 &pm; 1.91$ |
> - Yes, you are correct that ITES demonstrates an advantage in terms of reward but performs worse in terms of safety in the SafetyGymPointGoal1 and SafetyGymCarGoal1 environments due to the trade-off favoring performance. We address this issue in the first weakness discussion.
> - Yes, you are correct that ITES complicates the approach because it requires the training of an additional high-level policy. However, our results across all benchmarks demonstrate that ITES outperforms all single-policy methods in terms of performance and exhibits less variance in safety, as shown in Table 1 and Table 2. Moreover, when considering more complex environments such as AntMazeCshape, AntMazeWshape, and SafetyGymPointGoal1, our method does not compromise on safety.
> # Questions:
> - We conducted additional experiments in the SafeAntMazeWshape and PointGoal sparse environments.
> - In our experiments we obtained that ITES sacrifices some safety in short-horizon (SafetyGym) tasks in favor of performance. We provided results for the case where the agent have the highest performance while minimazing cost violations. During our research experiments we observed that increasing the role of safety reduces the performance, but even there it is not zero. Did we understand your question correctly?
> - Since ITES outperforms single-policy algorithms in simpler environments, we interpret this question as whether we can improve the performance or safety of our method by modifying its various components. One possible modification is to change the RL policies used at the high and low levels. Currently, TD3, an off-policy method, is employed because it is more sample-efficient than on-policy approaches, so there is little justification for replacing it with on-policy model-free methods. We could also consider replacing the hierarchical policy with the HIGL algorithm, which may be a potential direction for future work, as HIGL demonstrates better performance compared to HRAC. Additionally, incorporating an MPC controller as a low-level policy for planning in imagination, as implemented in the MBRCE approach, could enhance the safety of our method.

---

> > ### Comment · Reviewer_6mim · 2024-11-26
> >
> > Thanks for the author's response. My score will remian unchanged.

---

> > > ### Author Response · Authors · 2024-11-27
> > >
> > > Thank you for your response. We would be pleased to discuss what could enhance the quality of our work. What comments might we consider?

---

> > > > ### Author Response · Authors · 2024-12-02
> > > >
> > > > Additionally, we have introduced another benchmark: the S-shape long-horizon map, which illustrates the comparison of ITES with HRAC-LAG and TD3-LAG.
> > > >
> > > > | Env | Method | Final Success Rate | Final Cost |
> > > > | --------------- | --------------------- | --------------------- | ---------------- |
> > > > | AntMazeSshape  | ITES | $0.084 &pm; 0.01$| $181 &pm; 34.3$ |
> > > > | AntMazeSshape | HRAC-LAG| $0.075 &pm; 0.025$ | $230.1 &pm; 15.5$ |
> > > > | AntMazeSshape | TD3-L | $ 0 &pm; 0$ | $104.405 &pm; 7.09$ |
> > > >
> > > > The results indicate that TD3-LAG fails to complete the task, achieving a quality score of 0 while incurring the lowest cost, as it remains stationary. Furthermore, it is evident that ITES outperforms HRAC-LAG in terms of safety while maintaining comparable performance.

---

### Official Review · Reviewer_HTxN · 2024-11-03

**Soundness:** 3
**Presentation:** 3
**Contribution:** 2
**Rating:** 5
**Confidence:** 3

**Summary:**

This paper addresses the challenge of safe exploration in reinforcement learning. To balance performance with constraint satisfaction, the authors use a hierarchical architecture in which the high-level policy outputs safe subgoals and the low-level policy learns to reach these subgoals. Leveraging a learnable world model, the agent is able to fully consider safety before taking actions. The authors test the proposed ITES algorithm in the SafetyGym simulation environment, demonstrating its effectiveness in balancing safety and performance across both short-horizon and long-horizon tasks.

**Strengths:**

- Using the world model to imagine safety before executing actions is an interesting idea.
- The paper is well-organized, and the hierarchical structure is explained clearly.
- The experimental results demonstrate ITES’s effectiveness in balancing safety with performance across both short- and long-horizon tasks.

**Weaknesses:**

- The authors state, "ITES is the only one that combines model-based and hierarchy approach." However, the necessity of combining these two approaches is not demonstrated (for instance, if the model-based approach alone sufficiently improves safety, then a hierarchical structure may not be needed). Is this combination simply an “A+B” type of innovation?
- The experiments are limited to simple navigation tasks, raising questions about the method’s effectiveness in more complex, real-world tasks beyond navigation.
- The implementation codes are not provided.

**Questions:**

- Is this hierarchical structure, in which the high-level policy outputs subgoals, generally applicable, or is it only suited to navigation tasks? Would the high-level policy need to be redesigned for different types of tasks?
- How is the safety cost C_M specifically set? How are the probability labels obtained, especially in real-world tasks outside of simulations?
- How is the interval scale for subgoals determined?
- In Table 2, why is the Final Cost for ITES consistently the highest?
- There are some minor typos, including:
  - In line 116, `r_int` should be `r_in`.
  - The first occurrence of “HRAC” should be expanded in full.
  - The \beta coefficients in Equation (6) and their settings need explanation, as does Equation (9).
  - The quality of Figures 5 and 6 needs improvement; for example, in Figure 5, the shaded area is difficult to distinguish, and “Cost Rate” should not be underlined.
  - The format of the references needs to be standardized and unified.

---

> ### Author Response · Authors · 2024-11-25
>
> We sincerely thank you for your valuable commentaries and questions. We updated the article and uploaded the revised version. Considering the feedback from all the reviewers, we conducted additional experiments on SafeAntMazeWshape (ITES, HRAC, HRAC-L, TD3LAG), PointGoal sparse (ITES, TD3LAG, MBPPOL, SAC-L), PointGoal1 (TD3LAG), and CarGoal1 (TD3LAG).
> # Weaknesses:
> - In our work we mostly concentrated on long-horizon tasks where it is challenging to flat algorithms to find any solution. Generally, the model-based approach improves by leveraging predictive dynamics to anticipate and mitigate risky actions, but, on its own, it struggles with scalability and generalization to such complex tasks. The hierarchical structure complements this by enabling task decomposition, which simplifies decision-making and allows the agent to operate efficiently across varying temporal and spatial resolutions.
>
> We present two types of experiments.
> 1) The first demonstrates that model-based safety enhances safety in Hierarchical Reinforcement Learning (HRL). In ITES, the model-based component plays a critical role by evaluating the controller policy using imagined trajectories, specifically optimizing safety when achieving subgoals generated by the HRL component(high level policy). In Figure 7 we provide such experiments where HRAC-SafeSubgoals = HRL and ITES = HRL + model-based safety.
> 2) The second experiment shows that combining HRL and safety improves safety in model-based approaches. Model-based safety alone is insufficient for safely solving the task because it optimizes safety based on imagined scenarios over a limited horizon (10 steps ahead for SafetyGym, 15 steps for AntMaze). As the distance increases, the model of the world diverges. However, by generating subgoals at a short distance, Model-based safety effectively minimizes safety risks related to those subgoals. For experiments see Appendix A.
>
> | Env | Method | Final Reward | Final Cost |
> | ------------------- | ------------- | ---------------- | ---------------- |
> |   PointGoal1  | ITES | $21.42 \pm 1.14$| $40.46 &pm; 0.12$ |
> | PointGoal1 | TD3-imagine safety| $20.1 \pm 0.1$ | $75 \pm 10.4$ |
>
> - In our testing benchmarks, we primarily rely on existing state-of-the-art (SOTA) benchmarks to evaluate two key properties: safety (using the SafetyGym benchmark) and hierarchical capabilities (through long-horizon tasks, such as variations of Ant Mazes). For the future development of ITES, we will focus on a broader range of tasks, particularly in robotic manipulation. However, we would like to highlight that most existing safe reinforcement learning algorithms (MBPPOL, MBRCE) have been evaluated on only a limited subset of tasks from the SafetyGym benchmark.
>
> - We have added a link to the implementation of our algorithm.
> https://anonymous.4open.science/r/ITES-677D
>
> # Questions:
> - Hierarchical structure of ITES is applicable for other tasks, but we should define the mapping function $\phi$ from states to goals manually for every specific task. The architecture of the hierarchy is not need to be redesigned.
> - The cost model is a binary classifier that distinguishes safe and dangerous states. We define it as a MLP  approximator, and use Binary Cross Entropy as a loss function for learning. In the text, we added an additional description of defining and learning the cost model (Section 3.2). The safety states are task specific, so they are defined for each task to take into account safety constraints - this is like an engineering task, then to find a desirable solution we need to define a reward function, but here we also need a cost function. Defining cost and reward functions is a challenging problem that mostly depends on the exact task including real world tasks.
> - We would like to clarify your definition of "interval scale" in the context of subgoals. We interpret this to mean the number of low-level (primitive) controller steps that occur between subgoals identified by the high-level policy. In this framework, the interval scale is represented by a constant hyperparameter specific to each task. This hyperparameter is denoted as $k$ in the text, and its value is provided in Table 4 in the Appendix.
> - Indeed, you are correct. In Table 2, the average value of FinalCost for our method is greater than that of the other approaches. While we do have a higher average rate of constraint violations, it is important to emphasize that our method demonstrates greater stability. Specifically, the standard deviations are 89 for CUP and 93 for FOCOPS in the CarGoal environment, whereas our method, ITES, has a standard deviation of 1.04 for the Final Cost metric. This suggests that, for certain seeds, CUP and FOCOPS may exhibit excessive violations of safety constraints, whereas our method shows reduced sensitivity to seed variations, thereby enhancing its reliability.
> - Thank you for highlighting the typos; we have addressed them.

---

> > ### Author Response · Authors · 2024-11-27
> >
> > Thank you for the improved score. We have taken into account the comments of all the reviewers. Do you have any additional suggestions for enhancing the manuscript. Could you please share them?

---

> > > ### Author Response · Authors · 2024-12-02
> > >
> > > Additionally, we have introduced another benchmark: the S-shape long-horizon map, which illustrates the comparison of ITES with HRAC-LAG and TD3-LAG.
> > >
> > > | Env | Method | Final Success Rate | Final Cost |
> > > | --------------- | --------------------- | --------------------- | ---------------- |
> > > | AntMazeSshape  | ITES | $0.084 &pm; 0.01$| $181 &pm; 34.3$ |
> > > | AntMazeSshape | HRAC-LAG| $0.075 &pm; 0.025$ | $230.1 &pm; 15.5$ |
> > > | AntMazeSshape | TD3-L | $ 0 &pm; 0$ | $104.405 &pm; 7.09$ |
> > >
> > > The results indicate that TD3-LAG fails to complete the task, achieving a quality score of 0 while incurring the lowest cost, as it remains stationary. Furthermore, it is evident that ITES outperforms HRAC-LAG in terms of safety while maintaining comparable performance.

---

### Official Review · Reviewer_5JXL · 2024-11-04

**Soundness:** 2
**Presentation:** 3
**Contribution:** 2
**Rating:** 5
**Confidence:** 3

**Summary:**

This paper proposes a Safe RL algorithm that learns the model of the environment as well as a hierarchical approach to policy learning. In particular, it learns a high-level policy that creates sub-goals and a low-level policy that tries to achieve/reach the sub-goals. The authors claim improved results compared with Safe RL approaches that are based on primal-dual/Lagrangian relaxation.

**Strengths:**

- well-written and clear
- problem and approach is well-motivated

**Weaknesses:**

- There are not too many benchmarks on which this algorithm has been tested (only three)
- There are very few Safe RL baselines used for comparison. TD3Lag is the only primal-dual baseline, and it performs decently well but was used only in SafeAntMaze. Consider comparing with more primal-dual/Lagrangian relaxation approaches.
- It is not clear if any model-based safe RL algorithms were used as baselines, especially when the proposed approach is model-based. How does the proposed approach compare empirically with Jayant & Bhatnagar, 2022?
- It would be nice to have a pseudocode of the algorithm in one place.
- It would also be nice to have the training parameters used for the algorithm.

Minor errors:
- Section 2.1 title typo "constrained"

**Questions:**

What are the limitations of this approach?

---

> ### Author Response · Authors · 2024-11-22
>
> Thank you for valuable comments and constructive suggestions. We uploaded the updated version of the article. We introduced several additional benchmarks and conducted a set of new experiments on them. Also additional baseline algorithms were added. We improve the text (see highlighted zones in the text) according to your comments. Detailed description for each point follows below.
>
> # Weaknesses:
>
> - To enhance the number of used benchmarks we added SafeAntMaze (W shape) and GoalPoint sparse. These tasks are more complex than existing ones. The former contains the enlarged maze with modification in walls. The latter is the SafetyGym Goal Point, but the reward is dense: only +1 for achieving the goal. It is also worth mentioning that, compared to many works in the field of Safe Reinforcement Learning MBPPOL[1], MBRCE[2], SAC-L[3], CUP[4], FOCOPS[5], which evaluate their performance exclusively in SafetyGym navigation environments, we also consider long-horizon SafeAntMaze environments in addition to SafetyGym navigation.
>
> - To compare with other baseline algorithms we added SAC-L(lagrangian-based), MBPPOL(model-based).
> We compared our approach with MBPPOL in the PointGoal sparse environment and demonstrated that, with equivalent safety levels, ITES outperforms MBPPOL in terms of reward function. We did not compare MBPPOL in long-horizon AntMaze environments because its code cannot be reproduced in other environments or would require excessive time to run due to poor code quality.
>
> | Env | Method | Final Reward | Final Cost |
> | ------------------- | ------------- | ---------------- | ---------------- |
> |   PointGoal  sparse  | ITES | $8.64 &pm; 0.38$| $33.455 &pm; 1.47$ |
> | PointGoal  sparse | MBPPOL | $6.15 &pm; 0.15$ | $34.1 &pm; 3.6$ |
> | PointGoal  sparse | SAC-L| $0.675 &pm;0.08$ | $64.8 &pm; 3.8$ |
> | PointGoal  sparse | TD3-L | $ 0.07 &pm; 0.006$ | $64.7 &pm; 1.91$ |
>
>
> - We combined all training logic into pseudocode and provided the corresponding algorithm in the Appendix.
>
> - List of the hyperparameters was also added to the Appendix.
>
> - Thank you for noticing the typo, we fixed it.
>
>
> # Questions:
> - ITES has several limitations. The main limitation is the need to manually design the mapping function from state space to goal space for each task, which restricts its adaptability, particularly to visual input. For example, for 2D navigation tasks the goals are 2D coordinates, but for 3D navigation we should manually modify mapping to 3D coordinates.
>
> - The simple discretization technique used by HRAC's Adjacency Network limits its scalability to high-dimensional spaces.
>
> - ITES does not account for task-specific cost budgets, as it minimizes cost violations independently at each time step. This leads to minimization of the cost violations than the agent policy is already safe.
>
> - We updated text to highlight these limitations in the Conclusion Section.
>
>
> [1]Jayant, Ashish K., and Shalabh Bhatnagar. "Model-based safe deep reinforcement learning via a constrained proximal policy optimization algorithm." Advances in Neural Information Processing Systems 35 (2022): 24432-24445.
> [2]Liu, Zuxin, et al. "Constrained model-based reinforcement learning with robust cross-entropy method." arXiv preprint arXiv:2010.07968 (2020).
> [3] Yang, Qisong, et al. "WCSAC: Worst-case soft actor critic for safety-constrained reinforcement learning." Proceedings of the AAAI Conference on Artificial Intelligence. Vol. 35. No. 12. 2021.
> [4] Yang, Long, et al. "Cup: A conservative update policy algorithm for safe reinforcement learning." arXiv preprint arXiv:2202.07565 (2022).
> [5] Zhang, Yiming, Quan Vuong, and Keith Ross. "First order constrained optimization in policy space." Advances in Neural Information Processing Systems 33 (2020): 15338-15349.]

---

> > ### Comment · Reviewer_5JXL · 2024-11-26
> >
> > Thank you, I increase my score to 5

---

> > > ### Author Response · Authors · 2024-11-27
> > >
> > > Thank you for your response and for increasing the score. We would like to further enhance our work; could you please clarify any additional comments you may have that we could address? We invite you to review our most recent version of the manuscript, which takes into account all current feedback.

---

> > > > ### Author Response · Authors · 2024-12-02
> > > >
> > > > Additionally, we have introduced another benchmark: the S-shape long-horizon map, which illustrates the comparison of ITES with HRAC-LAG and TD3-LAG.
> > > >
> > > > | Env | Method | Final Success Rate | Final Cost |
> > > > | --------------- | --------------------- | --------------------- | ---------------- |
> > > > | AntMazeSshape  | ITES | $0.084 &pm; 0.01$| $181 &pm; 34.3$ |
> > > > | AntMazeSshape | HRAC-LAG| $0.075 &pm; 0.025$ | $230.1 &pm; 15.5$ |
> > > > | AntMazeSshape | TD3-L | $ 0 &pm; 0$ | $104.405 &pm; 7.09$ |
> > > >
> > > > The results indicate that TD3-LAG fails to complete the task, achieving a quality score of 0 while incurring the lowest cost, as it remains stationary. Furthermore, it is evident that ITES outperforms HRAC-LAG in terms of safety while maintaining comparable performance.

---

### Author Response · Authors · 2024-12-02
**General Response**

We would like to thank the reviewers for their positive feedback on our work and for their valuable suggestions, which we have tried to incorporate to improve the quality of our paper. In particular, we have focused on the following aspects:
- Investigating the contribution of each module (model-based, HRL) individually to the overall algorithm and providing source code implementation (HTxN);
- Comparing our approach with model-based methods, additional primal-dual baselines, and investigating a more complex Safety Gym environment and defining the applicability domain of the algorithm (5JXL);
- Analyze our method in the context of hierarchical baselines (nj6z);
- Consider additional long horizon environments (6mim);

In response to the feedback, we have implemented several improvements:
- Method: We have added details to the description of the cost model and provided information about the hyperparameters as well as the limitations of the proposed method.
- Experiments: We have added experiments in new environments - SafeAntMaze W-shape and GoalPoint sparse - along with their descriptions. We also added comparisons with TD3LAG in SafetyGym environments and included MBPPOL and SAC-L baselines.
- Appendix: We have added an appendix section that includes the pseudocode of the algorithm, a table of hyperparameters (Table 4), and a comparison with a single policy algorithm based on our proposed safety and our method.
- Text: We corrected typographical errors and improved the readability of the graphs.
- Code: We have added a link to the repository containing the source code: https://anonymous.4open.science/r/ITES-677D.
We are happy to answer any questions.

---

### Meta-Review · Area_Chair_JYpw · 2024-12-20

**Metareview:**

This paper proposes a model-based hierarchical RL algorithm for safe RL. The claim is that the method demonstrates superior performance on the benchmark while maintaining comparable safety violations.

### Strengths
Reviewers commented that the paper was well-written and clear,  the problem and approach is well-motivated, the idea was interesting, the paper is well-organized, the experimental results demonstrate the effectiveness in performance across both short- and long-horizon tasks, and the approach is practical.

### Weaknesses
Reviewers commented that the benchmark evaluation is limited, very few Safe RL baselines are included, that the method is an "A+B" type of innovation, the experiments are limited to simple navigation tasks, that the method is outperformed in simpler Safety Gym tasks, that the method only used one complex long-horizon task which restricts the evidence supporting ITES's generalizability to other challenging environments, that the evidence suggests it is less optimal for environments that strictly require safety prioritization, that the benefits of the added complexity are not consistently clear across all tasks, that the method is very similar, and that the paper does not address the limitations of model inaccuracies or present strategies to handle model uncertainty

The authors responded to the reviews, and then almost all of the reviewers engaged with the authors. The reviewers generally had reservations about the contribution of the method due to limited performance gains. This paper is borderline: while there is some evidence to support the claims, the results are somewhat mixed. One takeaway from the reviews is that this paper would be substantially improved with more comprehensive experiments. I recommend rejecting this paper in its current form.

**Additional Comments On Reviewer Discussion:**

No additional comments on reviewer discussion (see above metareview)

---

### Decision · Program_Chairs · 2025-01-22

Reject